# On Laws of Thought—A Quantum-like Machine Learning Approach

**DOI:** 10.3390/e25081213

**Published:** 2023-08-15

**Authors:** Lizhi Xin, Kevin Xin, Houwen Xin

**Affiliations:** 1Independent Researcher, Hefei 230026, China; lizhi_xin@yahoo.com; 2Independent Researcher, Chicago, IL 60607, USA; kevin_xin@yahoo.com; 3Department of Chemical Physics, University of Science and Technology of China, Hefei 230052, China

**Keywords:** quantum-like decision theory, genetic programming, machine learning, quantum gates, efficient market hypothesis

## Abstract

Incorporating insights from quantum theory, we propose a machine learning-based decision-making model, including a logic tree and a value tree; a genetic programming algorithm is applied to optimize both the logic tree and value tree. The logic tree and value tree together depict the entire decision-making process of a decision-maker. We applied this framework to the financial market, and a “machine economist” is developed to study a time series of the Dow Jones index. The “machine economist” will obtain a set of optimized strategies to maximize profits, and discover the efficient market hypothesis (random walk).

## 1. Introduction

We live in a world brimming with uncertainty, where we constantly have to make a lot of decisions with incomplete information. How we make decisions is truly an enigma. Classical decision theory [1,2,3,4,5] is a “black box”; we do not know what really happens inside the box. The human behaviors exhibited during decision making, such as the order effect, cannot be sufficiently explained by decision theory based on classical probability. Scientists are trying to apply quantum theory to reveal how decisions are made. Recently, many quantum-like decision theories [6,7,8] have been proposed based on quantum probability to revise the mathematical structure that is used in classical models. Aerts et al. first proposed to apply quantum probability in decision theory [9,10]; Busemeyer et al. proposed a quantum-like model to describe human judgments and the order effect [11,12,13]; Khrennikov et al. improved the Busemeyer quantum model by applying quantum instruments of quantum measurement theory [14,15,16,17,18]; Yukalov et al. proposed a rigorously axiomatic quantum decision theory [19,20,21]; Xin et al. proposed a quantum value operator decision theory [22].

Whether it is classical decision theories or quantum-like decision theories, all well-developed decision models have applied a rigorous mathematical structure to describe people’s decision making under uncertainty. We are firm believers that people’s subjective belief cannot be computed by rigorous mathematical formula. The main issue with mathematical models is that they are difficult to understand, cannot reflect the dynamic changes in the state of the decision-makers’ mind, and it is not easy to calculate theoretical values to compare with actual observed outcomes when the mathematical model becomes more complex.

In this paper, based on Darwin’s natural selection, we propose an algorithm that incorporates insights from quantum theory to describe people’s decision making under uncertainty. Our decision model emphasizes machine learning, where decision-makers build-up their experience by being rewarded or punished for each decision they make, preparing them to make better decisions in the future. This is more in line with decision-makers in the real world.

Our proposed quantum-like decision theory discovers laws of thought by machine learning an observed time series; there is no differential equation, and no transition probability computation in our decision theory. We do not model with the usual utility function or observables of the projection-type in other quantum-like decision theories, but a logic tree and a value tree. The logic tree determines the state of each point in the time series, and the value tree calculates the absolute value between two points in the time series. A logic tree and value tree work together to depict the entire decision-making process of a decision-maker.

In this paper, a “machine economist” is developed and the Dow Jones index is used as historical data for training the “machine economist”. The “machine economist” will trade Dow Jones index futures, build up experience, optimize a set of trading strategies to maximize profits, and finally construct a theory about financial markets.

## 2. Quantum-like Machine Learning Algorithm

The change in the Dow Jones index over time can be defined in terms of a time series consisting of states and observable values in (1).
(1a)$$\left\{\left(q_{k},x_{k}\right)\right\}k=1,\cdots ,N$$
(1b)$$q_{k}=\left\{\begin{array}{c} 0,\;\;x_{k}\geq x_{k-1}\\ 1,\;\;x_{k}<x_{k-1} \end{array}\right.$$
where $q_{k}$ denotes the dynamic state of the Dow Jones index; if the closing price of the Dow Jones index goes up, then the state is 0 ${(q}_{k}=0)$; if the closing price of the Dow Jones index goes down, then the state is 1 ${(q}_{k}=1)$; $x_{k}$ denotes the observed value (closing price) of the Dow Jones index; data sequences $\left\{x_{k},k=1,\cdots ,N\right\}$ describe the trajectory of the Dow Jones index.

Time series $\left\{\left(q_{k},x_{k}\right)\right\}$ can be considered as questions posed by the market, in which “machine economist” need to describe and interpret the market based on observed data sequences. The “machine economist” is actually playing with the market, and the market is neither optimistic nor pessimistic, but is just playing dice with the “machine economist”. The “machine economist” tries to maximize the expected value in order to find the most probabilistically correct answers (maximize profits).

The question now becomes: Can the “machine economist” find an answer?

Here, a quantum-like machine learning algorithm is proposed to answer the questions posed by the market. A quantum-like machine learning algorithm can be expressed in terms of two-part trees: the first part is a logic tree, which applies “yes or no” answers to determine the dynamically changing state of the security, and the other part is a value tree that describes the closing prices of the security. Together, the logic tree and value tree will reconstruct the trajectory of the security.

Logic tree: to determine the action to be taken and calculate the theoretical value of the closing price.Value tree: to calculate the absolute value of the difference in closing prices between two trading points of the Dow Jones index.

The goal of an algorithm $A_{k}$ in general is to be able to either:(1)Generate the results to match the observed outcomes;(2)Predict the next outcome.

In other words, given a sequence of data $\left\{\left(q_{k},x_{k}\right)\right\}$ as an input by the market, a “machine economist” develops an algorithm $A_{k}$ to output a sequence of data $\left\{\left(q_{k}^{\prime },x_{k}^{\prime }\right)\right\}$:(2a)$${\left\{\left(q_{k},x_{k}\right)\right\}\overset{input}{\to }A}_{k}\left(logicTree,valueTree\right)\overset{output}{\to }\left\{\left(q_{k}^{\prime },x_{k}^{\prime }\right)\right\}$$

Meet the following formula:(2b)$$q_{k}^{\prime }=q_{k}\;and\;x_{k}^{\prime }=x_{k},\;k=1,2,\cdots ,n$$
(2c)$$q_{n+1}^{\prime }=q_{n+1}\;and\;x_{n+1}^{\prime }=x_{n+1}$$

Genetic programming (GP) [23,24,25,26] is used by the “machine economist” to search for a satisfactory algorithm. Just as the genes of the fittest of each species are passed down from generation to generation through natural selection, evolution algorithms can perform the same action through machine learning.

The idea and steps of GP are simple:(1)Randomly generate 300 logic or value trees;(2)Historical data is learned to obtain the fitness of each tree;(3)The satisfactory logic or value tree is obtained through the Darwinian principle of survival of the fittest (crossover, mutation and selection) after about 80 generations of evolution.

The GP Algorithm (Algorithm 1) is as follows:

**Algorithm 1.** GP Algorithm

*Input:*
$\mathrm{Historical}\;\mathrm{dataset}\;\left\{\left(q_{k},x_{k}\right),k=0,\cdots ,N\right\}$ (each sample consists of a security’s state and closing price);Setting:
(1)Operation set F;(2)Dataset T;(3)Crossover probability = 70%; Mutation probability = 5%.

*Initialization:*
Population: randomly create 300 individuals.

*Evolution:*
Loop: for i=0 to 80 generations:
aCalculate fitness for each individual based on the historical dataset;bAccording to the quality of fitness:
iSelection: selecting parents.iiCrossover: generate a new offspring using the roulette algorithm based on crossover probability.iiiMutation: randomly modify the parent based on mutation probability.



*Output:*
An individual of the best fitness.



### 2.1. Value Tree

A value tree is a traditional function tree. The final form of the value tree is represented as a function. The output from this function is a numeric value. For a value tree the operation set F and dataset T are as follows
(1)Operation set $F=\left\{+,-,\times ,÷,log,exp\right\}$;(2)Dataset $T=\left\{t,fl,av,h,l\right\}.$
where t denotes the t-th trade of the Dow Jones index time series; fl denotes the average fluctuation of the closing price; av denotes the average closing price; h denotes the highest closing price; l denotes the lowest closing price.

A value tree is a function that consists of operation set F and dataset T.
(3)$$valueTree=f\left(F,T\right)$$

We define the absolute value of the Dow Jones index between two trading points as follows:(4)$$d_{t,t-1}=\left|x_{t}-x_{t-1}\right|$$

The “machine economist” can calculate the absolute value between two trading points using the value tree:(5)$$d_{t,t-1}^{\prime }=f\left(F,\left\{t,fl,av,h,l\right\}\right)-f\left(F,\left\{t-1,fl,av,h,l\right\}\right)$$

Now we can define the fitness function for the value tree as follows:(6)$${fitness}_{valueTree}=-\sum_{k=1}^{n}{\left(d_{t,t-1}^{\prime }-d_{t,t-1}\right)}^{2}$$
where $d_{t,t-1}^{\prime }$ is the absolute value calculated by the value tree in (5), and $d_{t,t-1}$ is the observed absolute value of the market in (4). The fitness function is essentially a particular type of function that is used to summarize, as a single figure of merit, how close a given design solution is to achieving the set aims. Fitness functions are used in GP to guide simulations towards optimal design solutions. In order to reach the optimal solution, the GP algorithm implements a continuous evolution process through selection, crossover, and mutation. The goal of continuous evolution is to find a satisfactory value tree that makes $d_{t,t-1}^{\prime }$ as close to $d_{t,t-1}$ as possible.

### 2.2. Logic Tree

A logic tree is a matrix tree constructed from eight basic quantum gates. The final form of a logic tree is represented as a matrix. The output from this matrix is a vector (an action, for example, buy or sell). The purpose of the logic tree is to simulate the decision-making process of the “machine economist”. Table 1 shows that the Dow Jones index has two states: $q_{1}$ (index up) and $q_{2}$ (index down); the “machine economist” has two possible actions to take, $a_{1}$ (buy) and $a_{2}$ (sell). $p_{1}\left|x,p_{1}\right|-x,p_{2}\left|-x,p_{2}\right|x$ are four possible outcomes determined by both the market and the “machine economist”. For example, $p_{1}|x$ means that the “machine economist” takes action $a_{1}$ (buy) with a subjective probability of $p_{1}$ and makes a profit x amount of money because the index is up ($q_{1}$); $p_{2}|-x$ indicates that the “machine economist” takes action $a_{2}$ (sell) with a subjective probability of $p_{2}$ and loses x amount of money because the index is up ($q_{1}$).

The market influences the traders’ decisions, while all traders’ actions then decide the market’s state. This interaction between the two, the objective (state of the market) and the subjective (traders’ beliefs), is what causes both the result of the decisions (gain or loss) and the state of the market (up or down) to be uncertain.

The state of the market describes the objective world; it can be represented by the superposition of all possible states in terms of the Hilbert state space as shown below [27,28].
(7)$$\left.|\psi \right\rangle =c_{1}\left.|q_{1}\right\rangle +c_{2}\left.|q_{2}\right\rangle$$
where $\left.|q_{1}\right\rangle$ denotes a state in which the market has increased, and $\left.|q_{2}\right\rangle$ denotes a state in which the market has decreased. ${\left|c_{1}\right|}^{2}$ is the objective frequency of the increase; ${\left|c_{2}\right|}^{2}$ is the objective frequency of the falling market.

The state of the trader’s mind is the subjective world. We postulate that when the trader is undecided in making a trade (buy or sell), it can be represented by superposition of all possible actions as follows.
(8)$$\left.|\phi \right\rangle =\mu_{1}\left.|a_{1}\right\rangle +\mu_{2}\left.|a_{2}\right\rangle$$
where $\left.|a_{1}\right\rangle$ denotes the trader’s action to buy, and $\left.|a_{2}\right\rangle$ denotes the trader’s action to sell. ${p_{1}=|\mu_{1}|}^{2}$ is the trader’s degree of belief in betting that the market will rise; ${p_{2}=|\mu_{2}|}^{2}$ is trader’s degree of belief in betting that the market will fall.

The information available to the “machine economist” prior to making its decision is incomplete; the “machine economist” does not know whether the market will rise or fall, forcing the “machine economist” to essentially guess. Before a “machine economist” makes a decision, its mind state is in a pure state, a superposed state in which it can decide whether to buy and sell at the same time. However, in reality, the “machine economist” cannot take an action to buy and sell simultaneously. This pure state is when the states of buy and sell are superposed in the “machine economist’s” mind. Then, when the “machine economist” makes the decision, the state of the “machine economist’s” mind then transforms from that pure state ρ into a mixed state $\rho^{\prime }$, which is when it decides to buy or sell, with certain degrees of belief. Basically, this transformation is the “machine economist” choosing from one of the available actions, with action a_1_ being buy with probability p_1_ and action a_2_ being sell with probability p_2_, shown below.
(9)$$Tradingprocess:\left.\rho =|\phi \right\rangle \left\langle \phi |\right.\overset{decision}{\to }\rho^{\prime }=p_{1}\left.|a_{1}\right\rangle \left\langle a_{1}|\right.+p_{2}\left.|a_{2}\right\rangle \left\langle a_{2}|\right.$$

It is expressed in matrix form as follows:(10a)$$\rho =\left[\begin{array}{cc} \rho_{11} & \rho_{12}\\ \rho_{21} & \rho_{22} \end{array}\right]\;\overset{diagonalization}{\to }\;\left[\begin{array}{cc} \lambda_{1} & 0\\ 0 & \lambda_{2} \end{array}\right]\overset{normalization}{\to }\rho^{’}=\left[\begin{array}{cc} p_{1} & 0\\ 0 & p_{2} \end{array}\right]\;=p_{1}\left.|a_{1}\right\rangle \left\langle a_{1}|+\right.p_{2}\left.|a_{2}\right\rangle \left\langle a_{2}|\right.$$
(10b)$$\left.|a_{1}\right\rangle =\left[\begin{array}{c} 1\\ 0 \end{array}\right],\;\left.|a_{2}\right\rangle =\left[\begin{array}{c} 0\\ 1 \end{array}\right];\;\left.|a_{1}\right\rangle \left\langle a_{1}|\right.=\left[\begin{array}{cc} 1 & 0\\ 0 & 0 \end{array}\right],\;\left.|a_{2}\right\rangle \left\langle a_{2}|\right.=\left[\begin{array}{cc} 0 & 0\\ 0 & 1 \end{array}\right]$$

The pure state (quantum density matrix) ρ can be approximately constructed from eight basic quantum gates. For a logic tree the operation set and dataset are as follows:
(1)Operation set $F=\left\{+,*,//\right\}$;(2)Dataset $T=\left\{H,X,Y,Z,S,D,T,I\right\}$
$$\left\{\begin{array}{c} \;H=\frac{1}{\sqrt{2}}\left[\begin{array}{cc} 1 & 1\\ 1 & -1 \end{array}\right]\;X=\left[\begin{array}{cc} 0 & 1\\ 1 & 0 \end{array}\right]\;Y=\left[\begin{array}{cc} 0 & -i\\ i & 0 \end{array}\right]\;Z=\left[\begin{array}{cc} 1 & 0\\ 0 & -1 \end{array}\right]\\ S=\left[\begin{array}{cc} 1 & 0\\ 0 & i \end{array}\right]\;D=\left[\begin{array}{cc} 0 & 1\\ -1 & 0 \end{array}\right]\;T=\left[\begin{array}{cc} 1 & 0\\ 0 & e^{i\pi /4} \end{array}\right]\;I=\left[\begin{array}{cc} 1 & 0\\ 0 & 1 \end{array}\right] \end{array}\right\}\;\;$$
where + means two matrices are added, ∗ means two matrices are multiplied, and // means that one is randomly selected from two branches. H,X,Y,Z,S,D,T,I are eight basic quantum gates (2 × 2 matrix) [29,30].

A logic tree is composed of operation set F and dataset T that determines the action taken by the “machine economist” and calculates the closing price of the Dow Jones index at the different trading points.
(11)$$logicTree=f\left(F,T\right)$$

With a logic tree the “machine economist” can decide the action to be taken $a_{t}$ and the closing price $x_{t}^{\prime }$ ($d_{t,t-1}^{\prime }$ can be calculated by the value tree in (5)):(12a)$$a_{t}=logicTree\left(F,T\right)=\left\{\begin{array}{c} 0,\;\;\mathrm{buy}\;\mathrm{is}\;\mathrm{excuted}\;(\mathrm{degrees}\;\mathrm{of}\;\mathrm{belief}\;\mathrm{is}\;p_{1})\\ 1,\;\;\mathrm{sell}\;\mathrm{is}\;\mathrm{excuted}\;(\mathrm{degrees}\;\mathrm{of}\;\mathrm{belief}\;\mathrm{is}\;p_{2}) \end{array}\right.$$
(12b)$$x_{t}^{\prime }=\left\{\begin{array}{c} x_{t-1}^{\prime }+d_{t,t-1}^{\prime },\;\;\mathrm{if}\;a_{t}=0)\\ x_{t-1}^{\prime }-d_{t,t-1}^{\prime },\;\;\mathrm{if}\;a_{t}=1) \end{array}\right.$$

The next step is to find a way to optimize the logic tree with a group of satisfactory strategies to guide the “machine economist’s” decisions. To optimize anything, there needs to be: first, a selection of a good evaluation function, and two, how to acquire an optimal solution. First off, in our model, the “machine economist” will try to maximize its expected value when making any trading decisions. Thus, we need to evaluate how “fit” the result (profit or deficit) of the “machine economist’s” decision are, which can be performed using the expected value in (13) as a fitness function to optimize the logic tree by evolving them. The whole idea of having GP go through an iterative evolution loop is to find a satisfactory logic tree by means of learning historical data to obtain the most optimal solution. The learning rules are as follows:(1)If the Dow Jones index is up ($q_{1}$):If the “machine economist” bets the Dow Jones index is up to buy ($a_{1}=0$), it profits $\left(d_{t,t-1}^{\prime }\right)$;If the “machine economist” bets the Dow Jones index is down to sell ($a_{2}=1$), it deficits $\left({-d}_{t,t-1}^{\prime }\right)$.(2)If the Dow Jones Index is down ($q_{2}$):
If the “machine economist” bets the Dow Jones index is down to sell ($a_{2}$ = 1), it profits $\left(d_{t,t-1}^{\prime }\right)$;If the “machine economist” bets the Dow Jones index is up to buy ($a_{1}=0$), it deficits $\left({-d}_{t,t-1}^{\prime }\right)$.


The expected t-th value of the “machine economist” is as follows:(13)$${EV}_{t}=\left\{\begin{array}{l} =p_{1}d_{t,t-1}^{\prime },\;\;\mathrm{market}\;\mathrm{is}\;\mathrm{up}\;\mathrm{and}\;\mathrm{the}\;“\mathrm{machine}\;\mathrm{economist}”\;\mathrm{buys}\;\mathrm{with}\;\mathrm{degrees}\;\mathrm{of}\;\mathrm{belief}p_{1}\;\;\\ ={-p}_{2}d_{t,t-1}^{\prime },\;\;\mathrm{market}\;\mathrm{is}\;\mathrm{up}\;\mathrm{and}\;\mathrm{the}\;“\mathrm{machine}\;\mathrm{economist}”\;\mathrm{sells}\;\mathrm{with}\;\mathrm{degrees}\;\mathrm{of}\;\mathrm{belief}p_{2}\\ ={-p}_{1}d_{t,t-1}^{\prime },\;\;\;\;\mathrm{market}\;\mathrm{is}\;\mathrm{down}\;\mathrm{and}\;\mathrm{the}\;“\mathrm{machine}\;\mathrm{economist}”\mathrm{buys}\;\mathrm{with}\;\mathrm{degrees}\;\mathrm{of}\;\mathrm{belief}p_{1}\\ =p_{2}d_{t,t-1}^{\prime },\;\;\mathrm{market}\;\mathrm{is}\;\mathrm{down}\;\mathrm{and}\;\mathrm{the}\;“\mathrm{machine}\;\mathrm{economist}”\;\mathrm{sells}\;\;\mathrm{with}\;\mathrm{degrees}\;\mathrm{of}\;\mathrm{belief}p_{2} \end{array}\right.$$

Now we can define the fitness function for the logic tree as follows:(14)$${fitness}_{logicTree}=\sum_{t=1}^{n}{EV}_{t}$$

${fitness}_{logicTree}$ maximizes “machine economist” expectations ($\underset{\mathrm{logicTree}}{\max}\left(\sum_{t=1}^{n}{EV}_{t}\right)$);${fitness}_{valueTree}$ applies negative feedback to make $\left(d_{t,t-1}^{\prime }\overset{equal}{\to }d_{t,t-1}\right)$. Logic tree together with value tree will reconstruct the trajectory of the Dow Jones index $\left\{\left(q_{k}^{\prime },x_{k}^{\prime }\right)\right\}\overset{equal}{\to }\left\{\left(q_{k},x_{k}\right)\right\}$, and make a prediction about the future outcomes as follows:(15a)$$d_{n+1,n}^{\prime }=valueTree\left(F,\left\{n+1,fl,av,h,l\right\}\right)-valueTree\left(F,\left\{n,fl,av,h,l\right\}\right)$$
(15b)$$a_{n}=logicTree(\left\{+,\times ,//\right\},\left\{H,X,Y,Z,S,D,T,I\right\})=\left\{\begin{array}{c} 0,\;\;“\mathrm{machine}\;\mathrm{economist}”\mathrm{takes}\;\mathrm{action}\;a_{1}(\mathrm{buy})\\ 1,\;\;“\mathrm{machine}\;\mathrm{economist}”\;\mathrm{takes}\;\mathrm{action}\;a_{2}(\mathrm{sell}) \end{array}\right.$$
(15c)$$x_{n+1}^{\prime }=\left\{\begin{array}{c} x_{n}^{\prime }+d_{n+1,n}^{\prime },\;\;\mathrm{if}\;a_{n}=0)\\ x_{n}^{\prime }-d_{n+1,n}^{\prime },\;\;\mathrm{if}\;a_{n}=1) \end{array}\right.$$

## 3. Results

The Dow Jones index from 1 to 30 December 2022 is used for training the “machine economist” as shown in Table 2. The first column indicates the state, where 0 means the index is up and 1 means the index is down; the second column indicates the closing price of the index. The first row is the initial condition, −1 indicates the state is uncertain, and 34,395.01 indicates the base closing price for machine learning.

### 3.1. Dow Jones Index’s Value Tree

By applying the fitness function of ${fitness}_{valueTree}$ (6), the “machine economist” can continuously learn the historical data to evolve a satisfactory value tree, as shown in Figure 1.
(16)$${valueTree}_{dowJones}=\left(t*fl\right)$$
where t denotes the t-th trade of the Dow Jones index, and fl denotes the average fluctuation of the closing price. The “machine economist” can use this value tree (${valueTree}_{dowJones}$) to calculate the absolute value between two trading points of the Dow Jones index:(17)$$d_{t,t-1}^{\prime }=t*fl-\left(t-1\right)*fl=fl=265$$

### 3.2. Dow Jones Index’s Logic Tree

By applying the fitness function ${fitness}_{logicTree}$ (14) and $d_{t,t-1}^{\prime }=265$(17), the “machine economist” can continuously learn the historical data of the Dow Jones index, and evolve a satisfactory logic tree as shown in Figure 2.

Figure 3 shows the optimization curve of the evolution algorithm. Figure 4 shows the learning curve of the evolution algorithm. As shown in Figure 4, the expected value of this logic tree is 5101, very close to the maximum expected value of the Dow Jones index time series (1–30 December 2022), which is 5308.

The ${logicTree}_{dowjones}$ (18) provides two strategies $\left\{S_{1}{,S}_{2}\right\}$, and the “machine economist” can randomly choose a strategy from the two and apply the strategy chosen to guide the “machine economist” in choosing which action to take (buy or sell with a subjective degrees of belief). If strategy $S_{1}$ is chosen, then the “machine economist” is 100% sure that the index is up (buy); if strategy $S_{2}$ is chosen, the “machine economist” is 100% sure that the index is down (sell). Combining ${logicTree}_{dowjones}$ (18) and $d_{t,t-1}^{\prime }$ (17), the “machine economist” can determine the action to be taken $a_{t}$ (19) and calculate the closing price $x_{t}^{\prime }$ (20) of the Dow Jones index.
(18)$${logicTree}_{dowjones}=\left(\left(D*\left(H+D\right)\right)*\left(\left(I//\left(Z+Z\right)\right)*I\right)\right)$$
$S_{1}=\left(\left(D*\left(H+D\right)\right)*\left(\left(Z+Z\right)*I\right)\right)\to \left.|a_{1}\right\rangle \left\langle a_{1}|\right.\left(p_{1}=100\%,p_{2}=0\right)$$S_{2}=\left(\left(D*\left(H+D\right)\right)*\left(I*I\right)\right)\to \left.|a_{2}\right\rangle \left\langle a_{2}|\right.\left(p_{1}=0,p_{2}=100\%\right)$




(19)
$$a_{t}=\left\{\begin{array}{c} \;0,\;\mathrm{if}\;S_{1}\;\mathrm{is}\;\mathrm{selected}\\ 1,\;\mathrm{if}\;S_{2}\;\mathrm{is}\;\mathrm{selected} \end{array}\right.$$


(20)
$$x_{t}^{\prime }=\left\{\begin{array}{c} x_{t-1}^{\prime }+265,\;\;\mathrm{if}\;a_{t}=0\;(\mathrm{buy})\\ x_{t-1}^{\prime }-265,\;\;if\;a_{t}=1\;(\mathrm{sell}) \end{array}\right.$$



By applying Equations (19) and (20), the “machine economist” can then reconstruct the trajectory of the Dow Jones index, as shown in Figure 5 with 19 wins and 1 loss. The ${logicTree}_{dowjones}$ randomly selects strategy $S_{1}$ and strategy $S_{2}$ with 9 buy events and 11 sell events, as shown in Figure 6, very close to the market 10 times up and 10 times down. In Figure 6, a positive bar represents a buy action taken by the logic tree, and 100 means with 100% degrees of belief to buy; a negative bar represents a sell action taken by the logic tree, and −100 means with 100% degrees of belief to sell. The random actions taken by ${logicTree}_{dowjones}$ to buy and sell imply that the evolution algorithm believes the market is efficient, i.e., the market is walking randomly.

Although the “machine economist” approximately reconstructs the price trajectory of the index, it can only make a 50/50 probability prediction of the future state of the Dow Jones index (up or down) by randomly choosing strategy 1 (believe the index is up) or strategy 2 (believe the index is down), i.e., without using partial differential equations and joint probabilities. The “machine economist” independently discovers that the market is efficient (random walk).

## 4. Discussion

More than a hundred years ago, Louis Bachelier found the similarity between stock price movement and Brownian motion by studying the Paris stock market data, and Bachelier applied a normal distribution to describe the movement in stock prices using stochastic differential equations. In this paper, the “machine economist” constructs a theory about the financial markets by studying a Dow Jones index time series, where the “machine economist” applies an algorithm and treats the data structure of the market as unknown, and the “machine economist” discovers that the market is efficient (random walk) through machine learning. It should be emphasized here that, unlike Bachelier, the “machine economist” discovers the efficient market hypothesis through machine learning with the evolution algorithm, without using any stochastic differential equations or rational economic man hypotheses.

Based on the superposition principle of quantum mechanics, we introduce objective (market) and subjective (trader) dual uncertainty to decision theory (Equations (7) and (8)). “Quantum jump” is applied to explain the decision process: that is, the decision process is a projection from a pure state to a mixed state (Equation (9)). A quantum density matrix in a pure state $\left.(\rho =|\phi \right\rangle \left\langle \phi |)\right.$ has quantum interference, such as Schrödinger’s cat who is dead and alive at the same time, a market that is up and down at the same time or a trader who can buy and sell at the same time; the mixed state is the classic statistical state, that is, the market can only be up or down or a trader can only buy or sell. Furthermore, we used eight fundamental quantum gates to construct a quantum density matrix (pure state), and optimized the quantum density matrix through evolutionary algorithms.

Time series in the complex real world rarely have a certain probability distribution, so the key is not to find the probability distribution from the time series but to find valuable information (experience or knowledge) and apply the learned experience to make decisions. In this paper, the “machine economist” uses both a logic tree and a value tree together to study historical data in order to obtain useful information (information of the state and the absolute value between two trading points). Instead of fitting a curve (price fluctuations of the Dow Jones index) with just one equation, the “machine economist” first uses the value tree to find the absolute value of the price difference between two trading points, and then uses the logic tree to determine the action to be taken (with degrees of belief). The value tree (objective) and logic tree (subjective) fit the curve together.

## Figures and Tables

**Figure 1 entropy-25-01213-f001:**
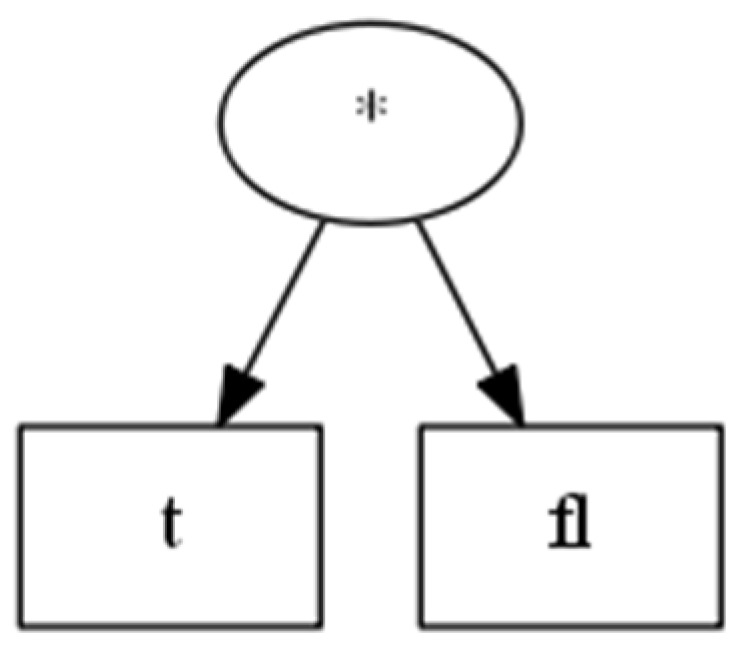
Value tree for the Dow Jones index.

**Figure 2 entropy-25-01213-f002:**
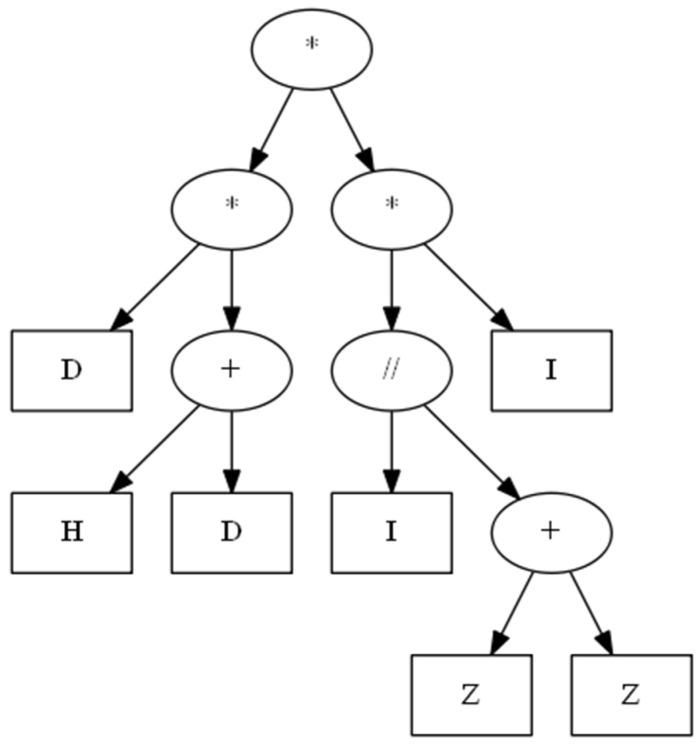
Logic tree for the Dow Jones index.

**Figure 3 entropy-25-01213-f003:**
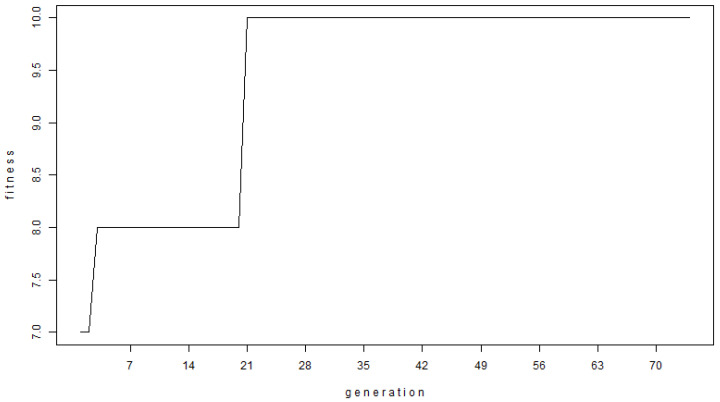
The optimization curve of the evolution algorithm.

**Figure 4 entropy-25-01213-f004:**
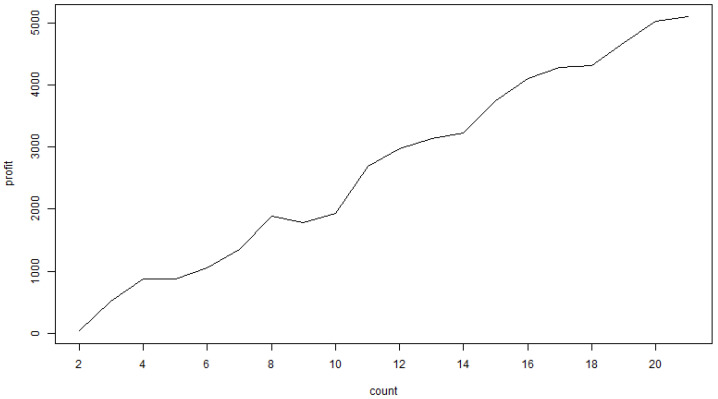
The learning curve of the evolution algorithm.

**Figure 5 entropy-25-01213-f005:**
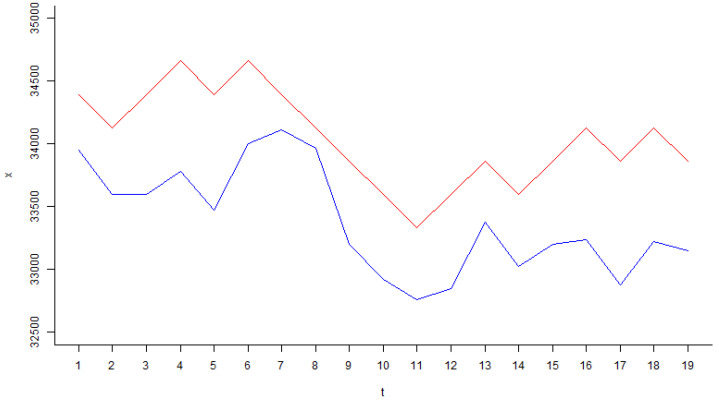
Trajectory of the Dow Jones index (blue line is the observed closing price, and the red line is the computed closing price).

**Figure 6 entropy-25-01213-f006:**
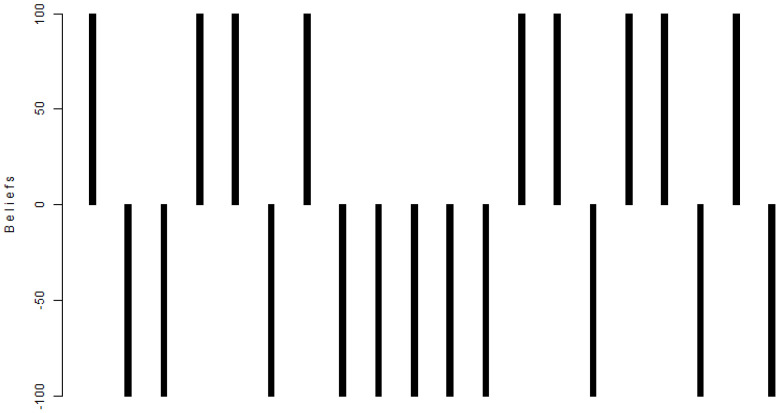
The buy (positive) or sell (negative) actions taken by the logic tree with degrees of belief (buy: 100; sell: −100).

**Table 1 entropy-25-01213-t001:** State–action–value decision table.

	State	$q_{1}$	$q_{2}$
Action			
$a_{1}$		$p_{1}|x$	$p_{1}|-x$
$a_{2}$		$p_{2}|-x$	$p_{2}|x$

**Table 2 entropy-25-01213-t002:** The Dow Jones index (1–30 December 2022).

−1	34,395.01
0	34,429.88
1	33,947.1
1	33,596.34
0	33,597.92
0	33,781.48
1	33,476.46
0	34,005.04
0	34,108.64
1	33,966.35
1	33,202.22
1	32,920.46
1	32,757.54
0	32,849.74
0	33,376.48
1	33,027.49
0	33,203.93
0	33,241.56
1	32,875.71
0	33,220.8
1	33,147.25

## Data Availability

The data that support the findings of this study are available on request.

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
