# Peer review of "On Laws of Thought—A Quantum-like Machine Learning Approach"

_entropy, 2023, doi:10.3390/e25081213_

Round 1

Reviewer 1 Report

1) The formal presentation of this paper is not good. Most equations are not placed in the center. 

2) Personally, I do not think too many abbreviations would benefit the reading experience, especially when the length of the paper is not limited.    

3) I am not following why the discussed algorithm is a quantum-like one since most techniques such as Eqs. (9)-(10) are nothing but linear algebra. I think the authors would give more explanations on the reason why it is a quantum-like or quantum-inspired one. By the way, I did not find the quotations in the paper beneficial to the understanding of the paper. I think the authors should further improve the presentation of the manuscript. 

The English writing of the manuscript is fine. 

Reviewer 2 Report

In this manuscript, the authors proposed a quantum-like machine learning algorithm for reproducing the coarse-grained trajectory of the Dow Jones index. The term “quantum-like” may refer to the superposition of the “state of the market” or “the state of machine trader” and the data set T. Using the proposed genetic programming algorithm therein, the author “reconstruct the price trajectory” but “can only make a 50/50 probability prediction”. The algorithm seems to make the overfitting on the available data using a logic decision tree and has no prediction capability. It is unclear whether the red line in Figure 3 is computed using a quantum computer. If the result in Figure 3 and Table 2 are not derived using quantum computing, defining the “state of the market,”  “the state of machine trader,” and the data set is meaningless.

In addition, the genetic programming algorithm in this manuscript is very similar to the proposed algorithm in the preprint with the arXiv number 2303.17607. In particular, the crossover probability and mutation probability in these two algorithms are set as the same as 70% and 5%, respectively. The contents and the algorithms in this manuscript and the preprint are very close to each. I am not convinced that the law of nature and “the law of thought” can be derived using  almost the same algorithm.

Reviewer 3 Report

The paper “On laws of thought - a quantum-like machine learning approach” is interesting, innovative, and can have some applications in predicting financial markets.

I have the following remarks on it.

1.      In spite of the title and general introduction given in wording of “quantum-like approach”, it actually has no relation to anything of “quantum-like” but rather presents a genetic algorithm approach to a machine learning elaboration on the prediction of the Dow-Jones index behavior.

2.      By this reason, the paper hardly fits the special issue on "Quantum Decision-Making: From Cognitive Psychology and Social Science to Artificial Intelligent Systems" but it still can be considered in the Entropy regular issue.

3.      For a regular issue, the paper can be adjusted by skipping the redundant “quantum” language and focusing directly on the genetic programming in machine learning.

4.      The genetic algorithms are pretty well known nowadays, so there is no need for giving two huge quotations on them in p. 3.

5.      Concerning application to stock market, a more substantial example than just one month data for December 2022 would be more reasonable.

6.      Estimation of the possible win in playing at the stock market during a chosen time period could be useful for demonstration of this approach predictive power.

7.     

can be improved.

Round 2

Reviewer 1 Report

I am ok with the revision

Reviewer 2 Report

According to the author's reply and the revised manuscript, few revisions have been made. Hence I do not recommend it for publication.

Reviewer 3 Report

The paper is improved in the revision, so it can be recommended for publishing.